# The Relationship between Students’ Physical Self-Concept and Their Physical Activity Levels and Sedentary Behavior: The Role of Students’ Motivation

**DOI:** 10.3390/ijerph18157775

**Published:** 2021-07-22

**Authors:** Juan J. Pulido, Miguel Ángel Tapia-Serrano, Jesús Díaz-García, José C. Ponce-Bordón, Miguel Á. López-Gajardo

**Affiliations:** Faculty of Sport Sciences, University of Extremadura, Av. de Elvsa, S/N, 10013 Caceres, Spain; matapiase@unex.es (M.Á.T.-S.); jdiaz@unex.es (J.D.-G.); jponcebo@gmail.com (J.C.P.-B.)

**Keywords:** adolescence, fitness, motivation, physical self-perceptions, sedentary behavior

## Abstract

This study aimed to analyze the association between specific dimension (i.e., fitness, appearance, physical competence, physical strength, and self-esteem) of students’ physical self-concept (PSC) and their physical activity (PA) levels (i.e., intentions to be physically active and out-of-school PA levels perceived by students) and sedentary behavior (SB) (i.e., total sitting and screen time perceived by students). We also tested the role of students’ motivation (i.e., intrinsic motivation and amotivation) towards PA in these relationships. A total of 1998 Spanish male (*n* = 1003) and female (*n* = 995) secondary students, aged between 13 and 17 years (*M* = 14.92, *SD* = 0.87) from 22 schools, enrolled in third grade (*n* = 1081) and fourth grade (*n* = 917), completed the self-reported questionnaires. Five independent structural equation modeling (SEM) adjusted by students’ sex was used to examine the association between specific dimension of students’ PSC (i.e., fitness, appearance, physical competence, physical strength, and self-esteem) and self-reported PA and SB variables, and to test the role of students’ motivation in these relationships. Overall, most of students’ PSC dimension positively predicted students’ PA outcomes, and were negatively associated with SB. In addition, most of students’ PSC dimension were positively associated with intrinsic motivation, and negatively predicted amotivation. In turn, intrinsic motivation was a positive predictor of PA outcomes, and a negative predictor of SB, whereas amotivation was a negative predictor of PA outcomes and sitting time, and a positive predictor of screen time. Finally, from PSC dimensions, sex as a covariate was a positive predictor of intrinsic motivation and PA outcomes, and was a negative predictor of amotivation and sitting time. These findings suggest the importance of the specific dimensions of PSC perceived by adolescents on their motivational processes, and in turn, on their PA and SB outcomes.

## 1. Introduction

Physical activity (PA) has been widely considered as a good health marker in adolescent students [1,2,3]. It is well-known that PA provides multiple health-related benefits in children and adolescents associated with intermediate outcomes (e.g., blood pressure, fitness, adiposity) and clinical outcomes (e.g., quality of life, cognition, mood) [4,5,6]. In addition, practicing PA continuously has been correlated with the decrease of several health and mental diseases (e.g., [7]). However, concern about inadequate levels of PA has recently emerged [8]. Prior research has demonstrated that sedentary behavior (SB) has increased from childhood to adolescence [9]. Many psychological backgrounds like a low students’ basic psychological needs degree (i.e., not satisfaction or need frustration) or a lack of motivation towards PA [10], and a poor physical self-concept (PSC) concerning the intensity of PA participation [11] could explain this situation. In this line, it would be interesting to extend the research area about students’ PSC, PA, and SB to promote healthy lifestyles.

SB has been studied in-depth in the last few years [12,13]. It is defined as any waking behavior characterized by an energy expenditure ≤ 1.5 METs while in a sitting or reclining position [13]. Despite a large number of guidelines to reduce SB [14,15], several studies have shown that SB has increased over the last years [12,13,14,15], which may lead to unfavorable measures of body composition, fitness, self-esteem, prosocial behavior, and academic achievement [16]. Research has reported that the main reasons for the increase of SB were associated with some sitting time behavior such as screen time [16], which has been associated with numerous mental health indicators, including hyperactivity/inattention problems, internalizing problems, and perceived quality of life [17], or (de)motivation towards PA.

Adherence to participating in PA has been arguably analyzed enough to prove that is closely related to students’ motivation [10,11]. Self-determination theory (SDT) [18] has been widely used to study students’ motivation towards PA. SDT distinguishes between intrinsic motivation, which is technically defined by activities performed “for their own sake”, or for their inherent interest and enjoyment [18]; extrinsic motivation, which assumes that an activity is performed for instrumental reasons [19], obtaining separable outcomes or to avoid disapproval; and amotivation, which refers to lacking intentionality [15]. Amotivation is very habitual in class-room settings and can be a result of either lack of felt competence to perform, or lack of the subject’s value or interest [19].

In the same vein of PA variables, PSC has been positively correlated with intrinsic motivation [10,11]. The reason for higher or lower PA levels may be associated with PSC perceived by adolescents [12]. Concretely, Marsh et al. [20] determined that PSC was composed by components related to health, coordination, activity, body fat, physical appearance, sport competence, global PSC, self-esteem, strength, flexibility, and endurance. More specifically, a global concept of PSC refers to a judgement a person has about their physical abilities when interacting with the environment [21]. PSC has been positively related to PA and negatively to SB [22]. Over last few years, research has shown that extended SB may cause diseases, regardless of PA practice levels [23]. Although sometimes SB can be positive (e.g., on a cognitive level, for instance, reading a book), most of the time is related to screen time (i.e., TV, computer, or smartphone; [24]). These PSC factors could explain why a person may adopt a more or less active level or SB. Students who perceive poor physical motor competence, self-esteem, or physical appearance may perceive PA as a barrier or mental limitation, PA rejecting any PA behavior and increasing their SB. Conversely, individuals who report high levels of PSC could also present higher PA levels and lower SB. That is, people who are satisfied with their look physically, have a good physical condition, feel competent in tasks that require motor physical involvement and, in turn, feel comfortable being physically active, which may increase their intention to be active during out-of-school leisure time and in the future [11,25].

### 1.1. Objectives

Objective 1. To analyze the relationship between specific dimensions of students’ PSC (i.e., fitness, appearance, motor/physical competence, physical strength, and self-esteem) with their PA levels (i.e., intentions to practice PA and out-of-school PA) and SB (i.e., sitting time and screen time).

Objective 2. To examine the role of students’ motivation (i.e., intrinsic motivation and amotivation) in the association with specific dimensions of students’ PSC (i.e., fitness, appearance, motor/physical competence, physical strength, and self-esteem) and with their PA levels and SB.

### 1.2. Hypothesis

**Hypothesis** **1.**
*Students’ PSC dimensions will be positively related to their (self-reported) PA levels (Hypothesis 1a), and negatively associated with their (self-reported) SB (Hypothesis 1b).*


**Hypothesis** **2.**
*Students’ PSC dimensions will be positively related to intrinsic motivation and negatively to amotivation (Hypothesis 2a). In addition, intrinsic motivation will be positively associated with students’ (self-reported) PA levels and negatively with their [self-reported] SB, whereas amotivation will be negatively related to students’ (self-reported) PA levels and positively with their [self-reported] SB (Hypothesis 2b).*


## 2. Materials and Methods

### 2.1. Participants and Procedures

The sample of the present study consisted of 1998 students (49.8% girls and 50.2% boys, mean age 14.92 years ± 0.87) from 22 Spanish schools, who filled out a set of questionnaires. Students were in 3rd (*n* = 1081) and 4th grade (*n* = 917). Class sizes ranged from 14 to 26 students per class. A total of 69 questionnaires (3.33%) were deleted for being more than 50% incomplete. A cross-sectional design was used. Previously, the main researcher contacted each school principal to explain the study’s objectives and to request their participation in the project. After they agreed to participate in the study, the research assistants provided each student with a letter of information about the voluntary participation and a parental consent form. Students’ who agreed to participate in the study completed paper-and-pencil questionnaires in their classrooms in a quiet environment and before a Physical Education lesson. The questionnaires were completed individually, typically within 15–20 min, supervised by the research assistants, and under non-distracting conditions.

### 2.2. Ethical Considerations

All research procedures were conducted following the Declaration of Helsinki [26], and the Ethical Committee of the University of Extremadura approved the study protocol (protocol code: 239/2019). Written informed consent was obtained from all participants who were involved in the study, according to the American Psychological Association’s ethical guidelines [27]. The data in this study were collected and analyzed anonymously.

### 2.3. Instruments

#### 2.3.1. Students’ Physical Self-Concept

PSC perceived by students was measured using the Spanish version [28] of the Physical Self-Perception Profile [29,30]. This scale contains 28 items corresponding to five factors that analyze: fitness (five items, e.g., “I always maintain an excellent physical condition and fitness”), appearance (six items, e.g., “I am very satisfied with how I look physically”), motor/physical competence (five items, e.g., “I am very good at almost every sport”), physical strength (five items, e.g., “I think that I am not as good as many others when I deal with situations in which strength is required”), and self-esteem (six-items, e.g., “I feel a little uncomfortable in places where physical exercise and sports are practiced”). The reliability (Cronbach’s alpha) of the instrument was tested for this study with acceptable [31] values of 0.80 for fitness, 0.70 for appearance, 0.82 for physical competence, 0.71 for physical strength, and 0.74 for self-esteem.

#### 2.3.2. Students’ Motivation towards PA

Students’ motivation towards PA was assessed using the Spanish version [32] of the Behavioural Regulation in Exercise Questionnaire (BREQ-3) [33]. Specifically, for the present study, we used the intrinsic motivation (four items, e.g., “because I feel pleasure and satisfaction when I do exercise”) and amotivation (four items, e.g., “I think I am wasting my time doing exercise”) factors. The scale presented an adequate internal consistency (Cronbach’s alpha) for the present study: 0.88 for intrinsic motivation and 0.86 for amotivation.

#### 2.3.3. Intentions to Practice PA

Students’ intention to practice PA was measured using a simple item [11]: “In the next future, I intend to participate in physical activity (at least twice a week)”.

All previous instruments were rated on a Likert scale ranging from 1 (*strongly disagree*) to 5 (*strongly agree*).

#### 2.3.4. Out-of-School PA Levels

Students’ out-of-school PA levels were assessed using a simple question: “How many hours do you dedicate to extracurricular physical activities (outside your class schedule) weekly?” The responses options ranged from 1 (*0 h*) to 6 (*more than 5 h*).

#### 2.3.5. Sitting Time

Students’ daily sitting time was measured using question number seven of the International Physical Activity Questionnaire (IPAQ) [34]. The specific question is: “During the last month, how long did you sit on one day of the week?” Students answered in “hours per day”, “minutes per day” and/or “I’m not sure”.

#### 2.3.6. Screen Time

Students’ daily screen time was assessed using the question [11]: “How many hours a day do you spend watching TV/tablet/smartphone or playing on the computer or video games?” Participants responded using a five-point Likert scale ranging from 0 (*0 h*) to 5 (*more than 5 five hours per day*). Other research had previously used single-item scales effectively [35,36].

### 2.4. Data Analysis

All statistical analyses were performed with Mplus version 7.3 [37]. In the main analyses, structural equation modeling (SEM) was used to test the relationship between variables with the following structure: (1) students’ PSC variables; (2) students’ intrinsic motivation and amotivation; and (3) students’ intentions to practice PA and out-of-school PA levels, and sitting time and screen time. More specifically, we tested five independent models, one by each specific dimensions of students’ PSC (i.e., fitness, appearance, motor/physical competence, physical strength, and self-esteem) on students’ intrinsic motivation and amotivation, and, in turn, on PA and SB outcomes. Previously, as preliminary analysis, an ANOVA was conducted including the students’ sex as a factor. We found significant differences in all dependent variables considering students’ sex. For this reason, we decided to include the sex as a covariate in our models.

## 3. Results

### 3.1. Descriptive Statistics and Bivariate Correlations

Table 1 shows the means, standard deviations, and bivariate correlations between variables. Concerning the correlations, specific dimensions of students’ PSC (except self-esteem) were positively related to intrinsic motivation. By contrast, students’ fitness and physical competence were negatively related to amotivation. Students’ intentions to practice PA was positively related to fitness, physical competence, and intrinsic motivation, and negatively associated with amotivation and SB outcomes. Out-of-school PA was also positively associated with all specific dimension of students’ PSC (except appearance). Finally, sitting time was negatively related to all specific dimension of students’ PSC and PA outcomes, and positively to screen time, whereas screen time was also negatively related to fitness, intrinsic motivation, and PA outcomes.

### 3.2. Main Analysis

Five independent SEM adjusted by students’ sex were used to examine the association between specific dimension of students’ PSC (i.e., fitness, appearance, physical competence, physical strength, and self-esteem) and self-reported PA and SB variables, and to test the role of students’ motivation in these relationships. Specifically, each dimension of students’ PSC was included as a predictor variable, intrinsic motivation, and amotivation towards PA as mediators, and intentions to practice PA, out-of-school PA levels, sitting time, and screen time as four criterion variables.

#### 3.2.1. From Students’ Fitness Perceptions to Intrinsic Motivation and Amotivation, and to Outcomes

Figure 1 shows the standardized results of the first model. Firstly, fitness was a positive predictor of intrinsic motivation (β = 0.57, *p* < 0.001), and a negatively predictor of amotivation (β = −0.36, *p* < 0.001). Secondly, intrinsic motivation positively predicted students’ intentions to practice PA (β = 0.16, *p* < 0.001) and out-of-school PA levels (β = 0.13, *p* < 0.001), and negatively predicted sitting time (β = −0.11, *p* < 0.01) and screen time (β = −0.07, *p* < 0.05). Additionally, amotivation negatively predicted students’ intentions to practice PA (β = −0.19, *p* < 0.001), out-of-school PA levels (β = −0.09, *p* < 0.01), and sitting time (β = −0.12, *p* < 0.01). Finally, fitness was a positive predictor of students’ intentions to practice PA (β = 0.20, *p* < 0.001) and out-of-school PA levels (β = 0.24, *p* < 0.001). The variance explained ranged between 1% (for screen time) to 36% (for intrinsic motivation; see *R*^2^*s* in Figure 1).

On the other hand, students’ sex included in the model as a covariate was a positive predictor of intrinsic motivation (β = 0.06, *p* < 0.01), intentions to practice PA (β = 0.05, *p* < 0.05), out-of-school PA levels (β = 0.13, *p* < 0.001), and a negative predictor of sitting time (β = −0.12, *p* < 0.001).

#### 3.2.2. From Students’ Appearance to Intrinsic Motivation and Amotivation, and to Outcomes

Figure 2 displays the standardized results of the second model. Firstly, appearance positively predicted intrinsic motivation (β = 0.10, *p* < 0.001), and negatively predicted amotivation (β = −0.05, *p* < 0.05). Secondly, intrinsic motivation positively predicted students’ intentions to practice PA (β = 0.28, *p* < 0.001) and out-of-school PA levels (β = 0.28, *p* < 0.001), whereas was a negative predictor of sitting time (β = −0.11, *p* < 0.01) and screen time (β = −0.09, *p* < 0.01). Additionally, amotivation negatively predicted students’ intentions to practice PA (β = −0.18, *p* < 0.001), out-of-school PA levels (β = −0.07, *p* < 0.01), and sitting time (β = −0.11, *p* < 0.01). Finally, appearance was a negative predictor of sitting time (β = −0.17, *p* < 0.001). The variance explained ranged between 1% (for amotivation or screen time) to 15% (for out-of-school PA levels; see *R*^2^*s* in Figure 2).

Including students’ sex in the model as a covariate, sex was a positive predictor of intrinsic motivation (β = 0.25, *p* < 0.001), intentions to practice PA (β = 0.09, *p* < 0.001), out-of-school PA levels (β = 0.18, *p* < 0.001), and a negative predictor of amotivation (β = −0.08, *p* < 0.001) and sitting time (β = −0.11, *p* < 0.001).

#### 3.2.3. From Students’ Physical Competence Perceptions to Intrinsic Motivation and Amotivation, and to Outcomes

Figure 3 shows the standardized results of the third model. Firstly, physical competence was a positive predictor of intrinsic motivation (β = 0.54, *p* < 0.001), and a negatively predictor of amotivation (β = −0.32, *p* < 0.001). Secondly, intrinsic motivation positively predicted students’ intentions to practice PA (β = 0.18, *p* < 0.001) and out-of-school PA levels (β = 0.18, *p* < 0.001), and negatively predicted sitting time (β = −0.10, *p* < 0.05) and screen time (β = −0.09, *p* < 0.01). Additionally, amotivation negatively predicted students’ intentions to practice PA (β = −0.19, *p* < 0.001), out-of-school PA levels (β = −0.10, *p* < 0.001), and sitting time (β = −0.11, *p* < 0.01). Finally, physical competence was a positive predictor of students’ intentions to practice PA (β = 0.18, *p* < 0.001) and out-of-school PA levels (β = 0.18, *p* < 0.001), and a negative predictor of screen time (β = −0.09, *p* < 0.01). The variance explained ranged between 1% (for screen time) to 30% (for intrinsic motivation; see *R*^2^*s* in Figure 3).

On the other hand, students’ sex included in the model as a covariate positively predicted out-of-school PA levels (β = 0.13, *p* < 0.001), whereas it negatively predicted sitting time (β = −0.12, *p* < 0.001) and screen time (β = −0.05, *p* < 0.05).

#### 3.2.4. From Students’ Physical Strength to Intrinsic Motivation and Amotivation, and to Outcomes

Figure 4 displays the standardized results of the fourth model. Firstly, physical strength positively predicted intrinsic motivation (β = 0.07, *p* < 0.01). Secondly, intrinsic motivation positively predicted students’ intentions to practice PA (β = 0.27, *p* < 0.001) and out-of-school PA levels (β = 0.18, *p* < 0.001), and negatively predicted sitting time (β = −0.12, *p* < 0.001) and screen time (β = −0.09, *p* < 0.01). Additionally, amotivation negatively predicted students’ intentions to practice PA (β = −0.18, *p* < 0.001), out-of-school PA levels (β = −0.09, *p* < 0.01), and sitting time (β = −0.11, *p* < 0.01). Finally, physical strength had a negative trend prediction on sitting time (β = −0.05, *p* = 0.052). The variance explained ranged between 1% (for amotivation or screen time) to 15% (for out-of-school PA levels; see *R*^2^*s* in Figure 4).

Including students’ sex in the model as a covariate, sex was a positive predictor of intrinsic motivation (β = 0.24 *p* < 0.001), intentions to practice PA (β = 0.09, *p* < 0.001), out-of-school PA levels (β = 0.18, *p* < 0.001), and a negative predictor of sitting time (β = −0.11, *p* < 0.001).

#### 3.2.5. From Students’ Self-Esteem to Intrinsic Motivation and Amotivation, and to Outcomes

Figure 5 shows the standardized results of the fifth model. Firstly, self-esteem was a negative predictor of intrinsic motivation (β = −0.07, *p* < 0.01). Secondly, intrinsic motivation positively predicted students’ intentions to practice PA (β = 0.27, *p* < 0.001) and out-of-school PA levels (β = 0.28, *p* < 0.001), and negatively predicted sitting time (β = −0.12, *p* < 0.001) and screen time (β = −0.09, *p* < 0.01). Additionally, amotivation negatively predicted students’ intentions to practice PA (β = −0.18, *p* < 0.001), out-of-school PA levels (β = −0.08, *p* < 0.01), and sitting time (β = −0.12, *p* < 0.01). Finally, self-esteem negatively predicted sitting time (β = −0.25, *p* < 0.001) and screen time (β = −0.06, *p* < 0.01). The variance explained ranged between 1% (for amotivation or screen time) to 15% (for out-of-school PA levels; see *R*^2^*s* in Figure 5).

On the other hand, students’ sex included in the model as a covariate positively predicted intentions to practice PA (β = 0.09, *p* < 0.001) and out-of-school PA levels (β = 0.18, *p* < 0.001), whereas negatively predicted amotivation (β = −0.07, *p* < 0.001) and sitting time (β = −0.13, *p* < 0.001).

## 4. Discussion

The purpose of the current study was to analyze the relationship between the specific factors of students’ PSC (i.e., fitness, appearance, physical competence, physical strength, and self-esteem) with students’ reported PA levels (i.e., intentions to practice PA and out-of-school PA) and SB (i.e., sitting time and screen time). Additionally, this study aimed to examine the role of students’ motivation (i.e., intrinsic motivation and amotivation) in the association with specific factors of students’ PSC and with students’ reported PA levels and SB. The results of each specific hypotheses as well as the implications, limitations, and potential future research directions related to this study are discussed in the remainder of this paper.

Firstly, Hypothesis 1 stated that specific dimensions of students’ PSC would be positively associated with their PA levels (i.e., intentions to practice PA and out-of-school PA; Hypothesis 1a), and negatively associated with their SB (i.e., sitting time and screen time; Hypothesis 1b). Hypothesis 1 was partially confirmed. Specifically, we found that students’ fitness and physical competence positively predicted students’ intentions to practice PA and out-of-school PA levels, whereas students’ appearance, physical strength, and self-esteem were negatively associated with sitting time, and students’ physical competence and self-esteem were also negatively related to screen time. Regarding Hypothesis 1a, adolescents who perceive a better fitness and they feel a good level of competence to carry out motor tasks, they reported a greater intention to practice PA and higher out-of-school PA levels. However, nonsignificant relationships were obtained between other dimensions of students’ PSC such as appearance, physical strength, or self-esteem and students’ self-report of higher or lower intentions to practice PA or out-of-school PA levels. A possible explanation of these findings could be that variables like fitness and physical competence can be more related to practice sport or physical activities. By contrast, dimensions such as appearance or self-esteem could be more associated with the students’ satisfaction of their physically look, regardless of whether adolescents are more active or less. These findings are in line with previous studies, which showed that adolescents with a better general PSC reported higher PA levels [12,38,39,40]. Although this study did not investigate the causes that may lead to these relationship based on an experimental design or and using objectively measured (i.e., PA and SB), it is well known that PSC could be specially related to perceived competence or fitness, especially during adolescence, which may, in turn, have a positive relationship with intentions to practice PA [12,38,39,40,41]. Therefore, adolescents with a better PSC could feel more physically competent and, in turn, have stronger intentions to practice PA and report higher levels of out-of-school PA. Increasing students’ physical competence or improving their fitness may be efficient as a strategy for increasing PA levels in young people but these findings should be carefully considered. Thus, more experimental studies are required.

Concerning Hypothesis 1b, only students’ PSC dimensions of appearance and self-esteem were negatively associated with sitting time. In other words, students who report better levels in their physical appearance and self-esteem they also perceive sitting activities for less time. Previous research has also shown a negative relationship between a general PSC and SB. It is known that adolescents with lower PSC may present poor perceived competence in motor tasks, and, in turn, the levels of sitting time may be higher [12,41]. However, the dimensions of PSC related to sitting time in our study are variables more associated with students’ appearance and self-esteem. The model proposed by Stodden et al. [41] describes adolescents with a sedentary lifestyle they have also low levels of general PSC. As perceived PSC decreases, the sense of physical appearance or self-esteem also decrease, entering a negative spiral of disengagement with PA that contributes to increasing sedentary time [41]. However, our results should be interpreted with caution and further research is necessary to confirm the relationship between PSC and sitting time and screen time perceived by students or objectively measured using accelerometry.

Secondly, Hypothesis 2 suggested that students’ variables of PSC will be positively related to intrinsic motivation and negatively to amotivation (Hypothesis 2a), and, in turn, students’ intrinsic motivation will be positively associated with students’ ( self-reported) PA levels and negatively with their (self-reported) SB. Additionally, amotivation will be negatively related to students’ (self-reported) PA levels and positively with their (self-reported) SB (Hypothesis 2b). In this regard, students’ fitness, appearance, physical competence, and physical strength were positively associated with intrinsic motivation, whereas students’ self-esteem negatively predicted amotivation. Intrinsic motivation was a positive predictor of PA outcomes, and a negative predictor of SB, whereas amotivation was a negative predictor of PA outcomes and sitting time, and a positive predictor of screen time. These findings suggest that when students are satisfied with their physically look, competent, capable and fit to do physical activities, they also feel a sense of pleasure when they perform physical activities. In addition, when students are intrinsically motivated, they reported higher intentions to practice PA and higher out-of-school PA levels. It is well known that intrinsic motivation towards sports practice plays an important role in the intention to practice PA and out-of-school PA levels, determining the initiation, continuation, and drop-out of PA [10,42]. Motivation is the main key to regulate people’s behaviors and PSC is positively related to intrinsic motivation, which, in turn, has a positive association on intention to practice PA and out-of-school PA levels. It is well known that behaviors adhering to higher intrinsic motivation are associated with positive behaviors, such as satisfaction and enjoyment, consequently improve competence, autonomy, and social relations [35]. Conversely, the negative relationship between PSC (i.e., fitness, appearance, and physical competence) and amotivation, which had a negative relation to sitting time, could be explained because amotivation denotes emotional states opposite to intrinsic motivation: decreased physical competence, autonomy, and social relations, and as a consequence, unhealthy behavior (e.g., sitting time; [43,44]). However, further exploration would be needed to analyze the positive relationship between amotivation and sitting time, although it can be that a people report high levels of PA and SB at the same time [45]. In other words, it maybe that amotivation towards PA does not necessarily associated with be more sedentary or less. Previous studies grounded in SDT have suggested that feelings of physical competence and autonomy are closely related to intrinsic motivation and amotivation. These feelings of an optimal and general PSC, therefore, can play a key role in adolescents’ intrinsic motivation and amotivation, consequently modifying individuals’ behaviors, for example, PA levels and SB [40,43]. Despite the current research did not measure extrinsic regulations in line of a previous study [11], further investigations should examine how other students’ motivations towards PA could be associated with PSC and outcomes related to students’ PA levels and SB. For instance, practicing PA by aesthetic reasons could mediate the relationship between PSC factors (i.e., appearance, self-esteem, or physical fitness among others) and students’ PA and SB levels.

Finally, as a complementary finding regarding students’ sex, in general, sex as a covariate was a positive predictor of intrinsic motivation from fitness, appearance, and physical strength, and PA outcomes from fitness, appearance, physical competence, and self-esteem. Additionally, sex was a negative predictor of amotivation from fitness, appearance, and self-esteem, and sitting time. These results suggest that boys feel more pleasure towards physical activities and report higher levels of PA outcomes than girls. Accordingly, girls are more amotivated to practice physical activities and mainly present higher levels of sitting time than boys. These findings are in line with numerous studies that have demonstrated relationship between variables related to PSC and PA outcomes. For instance, positive associations between fitness and total PA (i.e., using accelerometry) were found in boys but not in girls [46,47] in early adolescence [48]. Therefore, these sex differences can lead teachers to promote specific strategies to optimize the specific dimension of PSC, especially in girls, and their motivational processes, with the aim of increasing their PA levels and reducing sitting time.

Drawing on the findings of this study, recommendations or practical proposals can be made for implementation in a real context. On the one hand, our results could have relevant practical implications for enhancing PA in adolescence because PA levels are alarmingly low, according to recent reports [49]. This could include incorporating strategies to promote and favor the students’ PSC, which could help to relativize their body image [11] achieve better PA levels and reduce their sedentary lifestyle—doing so can lead to better health [50]. In addition to improving the PSC, our findings concerning the importance of students’ motivation suggest that teachers and practitioners should improve the students’ interest in and enjoyment of PA. This could be accomplished by utilizing some specific strategies, such as developing activities in which the students’ preferences are considered, allowing and facilitating their involvement considering also the sex characteristics. This may be particularly beneficial for those students who do not feel any interest or who lack motivation for PA. Otherwise, participation in sport activities could have an adverse effect (e.g., with higher amotivation values) and increase SB, such as spending more time sitting during the day.

Despite the study’s contribution to our knowledge of the relationship of students’ specific dimensions of PSC and motivation and PA levels and SB, there are some limitations that should be considered. First, this study used a cross-sectional design. Future longitudinal or experimental studies would allow us to examine changes in the identified relationships over time and provide more robust evidence of the relationship between PSC, motivational processes, and PA or SB. Second, the data of this research were collected with student self-reported questionnaires, which may imply social desirability bias. Therefore, in further works, it would be important to analyze the association of these variables using objective measures (e.g., accelerometers). In this regard, our findings are based on correlational designs, thus preventing the establishment of possible causal relationships. Despite significant improvements have been done measuring PA and SB levels using accelerometry, an established standard for the measurement of PA does not exist due to the complexity of the behavior [51]. PA is multifaceted and encompasses different domains (e.g., leisure, occupation, transport, household), dimensions (e.g., frequency, duration, intensity). Additionally, some instruments used in the current work present a single-item structure (e.g., screen time); therefore, it is not possible to test the validity and reliability of these scales. Third, as the authors only measured intrinsic motivation and amotivation, it was not possible to know how external regulations may mediate between PSC and PA levels and SB. It would be interesting for further investigations including the entire *spectrum* of motivational regulations (also measuring extrinsic regulations) to a better understanding in how PSC could be related to other students’ motivational regulations, their PA levels and SB. Finally, we have used an own created scale for the current study to assess out-of-school PA levels instead of using other previous validated instruments [52].

## 5. Conclusions

This research highlights the relationship between PSC and PA levels (i.e., intentions to practice PA and out-of-school PA) and sedentary behaviors (i.e., sitting time and screen time) in students. In particular, these results mainly suggest that most of students’ PSC variables are associated with more active and less SB. Moreover, we can conclude that students’ motivation plays an important role in the relationship with students’ PSC dimensions and, additionally, with their intentions to practice PA, out-of-school PA, and sitting time. Finally, sex differences should be considered in the relationship of these variables.

## Figures and Tables

**Figure 1 ijerph-18-07775-f001:**
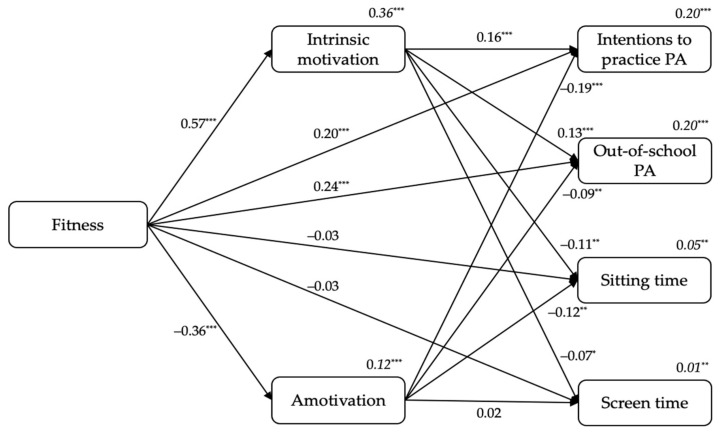
The standardized coefficients from students’ fitness to intrinsic motivation and amotivation, and to outcomes. *Note*. ^*^
*p* < 0.05. ^**^
*p* < 0.01. ^***^
*p* < 0.001.

**Figure 2 ijerph-18-07775-f002:**
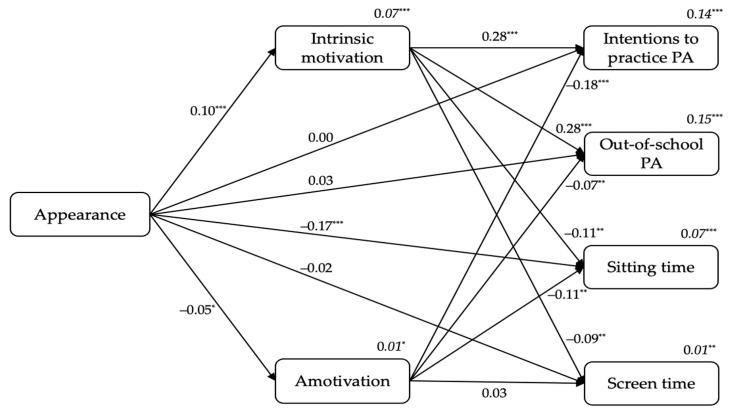
The standardized coefficients from students’ appearance to intrinsic motivation and amotivation, and to outcomes. *Note*. ^*^
*p* < 0.05. ^**^
*p* < 0.01. ^***^
*p* < 0.001.

**Figure 3 ijerph-18-07775-f003:**
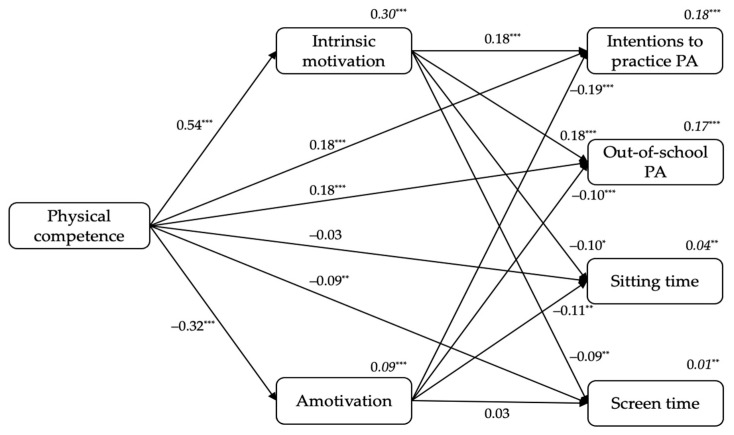
The standardized coefficients from students’ physical competence to intrinsic motivation and amotivation, and to outcomes. *Note*. ^*^
*p* < 0.05. ^**^
*p* < 0.01. ^***^
*p* < 0.001.

**Figure 4 ijerph-18-07775-f004:**
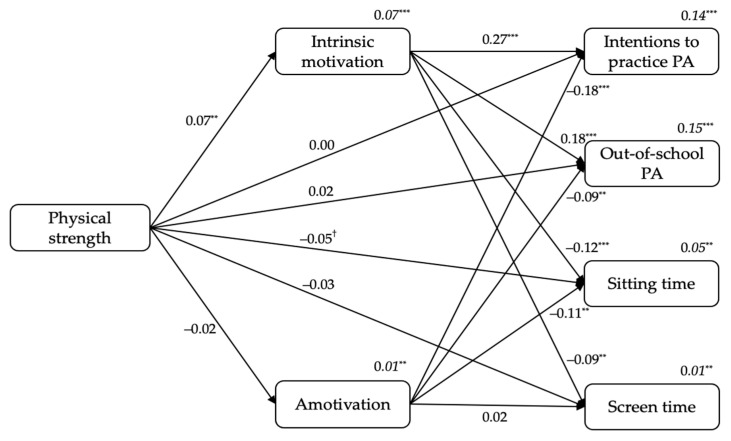
The standardized coefficients from students’ physical strength to intrinsic motivation and amotivation, and to outcomes. *Note*. ^†^
*p* < 0.05. ^**^
*p* < 0.01. ^***^
*p* < 0.001.

**Figure 5 ijerph-18-07775-f005:**
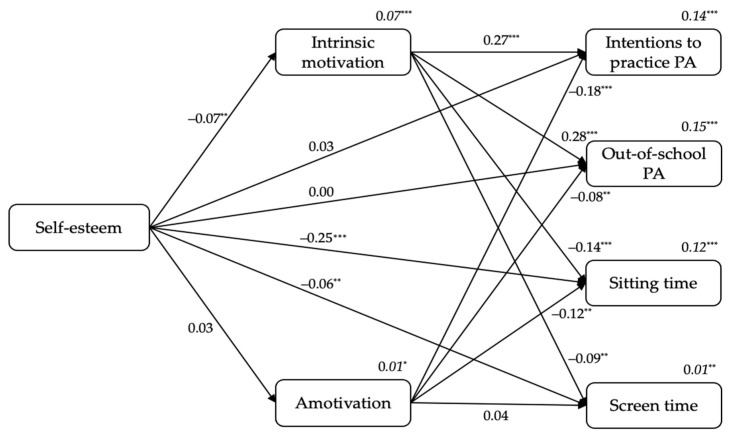
The standardized coefficients from students’ self-esteem to intrinsic motivation and amotivation, and to outcomes. *Note*. ^*^
*p* < 0.05. ^**^
*p* < 0.01. ^***^
*p* < 0.001.

**Table 1 ijerph-18-07775-t001:** Descriptive statistics and bivariate correlations between study variables.

Variables	*M* (SD)	1	2	3	4	5	6	7	8	9	10	11
1. Fitness	3.36 (0.91)	-					-					
2. Appearance	3.09 (0.67)	0.20 ^**^	-									
3. Physical competence	3.12 (0.89)	0.76 ^**^	0.17 ^**^	-								
4. Physical strength	2.89 (0.65)	0.19 ^**^	0.43 ^**^	0.24 ^**^	-							
5. Self-esteem	2.94 (0.98)	−0.07	0.60 ^**^	−0.08 ^*^	0.34 ^**^	-						
6. Intrinsic motivation	4.00 (0.97)	0.59 ^**^	0.11 ^**^	0.55 ^**^	0.11 ^**^	−0.07	-					
7. Amotivation	1.55 (0.84)	−0.34 ^**^	−0.05	−0.29 ^**^	−0.04	0.04	−0.66 ^**^	-				
8. Intentions to practice PA	4.33 (0.96)	0.37 ^**^	−0.05	0.34 ^**^	0.06	−0.00	0.41 ^**^	−0.36 ^**^	-			
9. Out-of-school PA (hours/week)	2.22 (1.73)	0.40 ^**^	0.08 ^**^	0.35 ^**^	0.09 ^**^	−0.02	0.38 ^**^	−0.28 ^**^	0.36 ^**^	-		
10. Sitting time (min/day)	452.17 (189.87)	−0.10 ^**^	−0.18 ^**^	−0.11 ^**^	−0.08 ^*^	−0.25 ^**^	−0.08 ^**^	−0.02	−0.05 ^*^	−0.04	-	
11. Screen time (hours/day)	2.30 (1.25)	−0.09 ^**^	−0.04	−0.06	−0.05	−0.05	−0.12 ^**^	0.09 ^**^	−0.13 ^**^	−0.07 ^**^	0.21 ^**^	-

Note. PA = physical activity, ^*^
*p* < 0.05, ^**^
*p* < 0.01.

## Data Availability

Data are available on https://osf.io/tjgue/ (accessed on 26 April 2021).

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
