# Peer review of "The Relationship between Students’ Physical Self-Concept and Their Physical Activity Levels and Sedentary Behavior: The Role of Students’ Motivation"

_ijerph, 2021, doi:10.3390/ijerph18157775_

Round 1

Reviewer 1 Report

I appreciate the efforts of the authors in addressing my concerns and those of the editor and other reviewers. The manuscript has been substantially improved as a result of the changes that have been made. I have no further comments on the manuscript at this time.

Author Response

I appreciate the efforts of the authors in addressing my concerns and those of the editor and other reviewers. The manuscript has been substantially improved as a result of the changes that have been made. I have no further comments on the manuscript at this time.

            Response: Thank you so much for your positive comments provided to our work.

Reviewer 2 Report

Dear Authors,

Congratulations! Now your manuscript seems to be ready to be published from my point of view.

There are many valuable changes which made the work sufficient. The only thing is with limitations which from my perspective should be more highlighted and justified for the Readers.

Author Response

Dear Authors,

Congratulations! Now your manuscript seems to be ready to be published from my point of view.

There are many valuable changes which made the work sufficient. The only thing is with limitations which from my perspective should be more highlighted and justified for the Readers.

Response: Thank you so much for your positive comments provided to our work. I have attended the suggestion about the limitations section, and we have accordingly done some modifications.

Reviewer 3 Report

Overall, the manuscript is much improved and with the new data analysis reported I have fewer concerns. 

Author Response

Reviewer 3

Overall, the manuscript is much improved and with the new data analysis reported I have fewer concerns.

Response: Thank you so much for your positive comments provided to our work.

This manuscript is a resubmission of an earlier submission. The following is a list of the peer review reports and author responses from that submission.

Round 1

Reviewer 1 Report

The authors ask an interesting question and should be commended for the number of participants that they were able to sample for this study. However, the introduction needs to build the rationale and there are some concerns about the use and testing of multiple models. Below are some more specific comments that may help strengthen the paper.

Comments

Line 31; can you be more specific on what you mean by health components here.

Lines 35-38; these concepts need to be developed a bit more. What does this research mean?

Line 49; what is meant by practicing PA? this sentence is a bit unclear.

Lines 58- 74; the integration of the concept of self-concept is a bit confusing. In one place they say it has 4 subdomains and then later talks about physical self-concept. It maybe clearer for the reader if the discussion of physical self-concept and its importance comes sooner.

Lines 76-81; the introduction needs to do a better job of helping support these objectives. Specifically, it is unclear based on the literature why self-concept would predict PA and SB. Additionally, new concepts of intentions and out-of-school PA are newly introduced. In the objective 2, why is extrinsic motivation not explored as well? This needs to be clearer.

 Line 99; what is meant by conditioned classrooms?

Line 110; is this supposed to measure the Marsh model; if so why doesn’t it examine all the subcomponents.

Lines 121-129; once again, why only these times of the BREQ-3?

Lines 129-137; please try to show these items in the introduction.

Data analysis; why did the authors create four models, it seems it could be just one model with each of the dependent variables all being together as dependent variables. This would help highlight what variables are of most importance and allow for a stronger discussion section to be written.

Figure 1 is very hard to read; can a higher quality figure be included?

Line 324-327; why was this the case; the authors really need to discuss at some point.

Final question; were any analyses performed to see if there were sex/gender differences in the variables or fit of the model?

Author Response

Letter of responses

Reviewer 1

The authors ask an interesting question and should be commended for the number of participants that they were able to sample for this study. However, the introduction needs to build the rationale and there are some concerns about the use and testing of multiple models. Below are some more specific comments that may help strengthen the paper.

Response: We would like to thank to the Reviewer 1 for the positive and valuable comments provided to our work. We have addressed all these recommendations point-by-point (i.e., marked in red color through the manuscript) with the aim to improve the quality of the manuscript.

Comments

Line 31; can you be more specific on what you mean by health components here.

Response: Thanks for this comment. We have rewritten the sentence to be more specific.

“It is well-known that PA provides multiple health-related benefits in children and adolescents associated with intermediate outcomes (e.g., blood pressure, fitness, adiposity) and clinical outcomes (e.g., quality of life, cognition, mood) [4-6]”.

Chaput, J.P.; Willumsen, J.; Bull, F.; Chou, R.; Ekelund, U.; Firth, J.; ... Katzmarzyk, PT. 2020 WHO guidelines on physical activity and sedentary behaviour for children and adolescents aged 5–17 years: summary of the evidence. Int. J. Behav. Nutr. Phys. Act. 2020, 17, 1-9.

Rhodes, R.E.; Janssen, I.; Bredin, S.S.; Warburton, D.E.; Bauman, A. Physical activity: Health impact, prevalence, correlates and interventions. Psychol. Health 2017, 32, 942-975.

Verswijveren S, Lamb KE, Bell LA, Timperio A, Salmon J, Ridgers ND. Associations between activity patterns and cardio-metabolic risk factors in children and adolescents: a systematic review. PLoS One. 2018; 13(8): e0201947.

Lines 35-38; these concepts need to be developed a bit more. What does this research mean?

Response: We have included more information to explain what these studies contribute.

“Many psychological backgrounds like a low students’ basic psychological needs degree (i.e., not satisfaction or need frustration) or a lack of motivation towards PA [10], and a poor physical self-concept (PSC) concerning the intensity of PA participation [11] could explain this situation”.

We have used PSC as abbreviation of physical self-concept according to another reviewer’s suggestion.

Line 49; what is meant by practicing PA? this sentence is a bit unclear.

Response: We have modified the beginning of the sentence as follow:

“Adherence to participating in PA has been arguably…”.

Lines 58- 74; the integration of the concept of self-concept is a bit confusing. In one place they say it has 4 subdomains and then later talks about physical self-concept. It maybe clearer for the reader if the discussion of physical self-concept and its importance comes sooner.

Response: Thanks again to the reviewer for this objection. We have modified the part of Introduction section about physical self-concept.

“In the same vein of PA variables, PSC has been positively correlated with intrinsic motivation [10-11]. The reason for higher or lower PA levels may be associated with PSC perceived by adolescents [12]. Concretely, Marsh et al. [20] determined that PSC was composed by components related to health, coordination, activity, body fat, physical appearance, sport competence, global PSC, self-esteem, strength, flexibility, and endurance. More specifically, a global concept of PSC refers to a judgement a person has about their physical abilities when interacting with the environment [21]. PSC has been positively related to PA and negatively to SB [22]. Over last few years, research has shown that extended SB may cause diseases, regardless of PA practice levels [23]. Although sometimes SB can be positive (e.g., on a cognitive level, for instance, reading a book), most of the time is related to screen time (i.e., TV, computer, or smartphone; [24]). These PSC factors could explain why a person may be more or less active or sedentary. Students who perceive poor physical motor competence, self-esteem, or physical appearance may perceive PA as a barrier or mental limitation, PA rejecting any PA behavior and increasing their SB. Conversely, individuals who report high levels of PSC could report higher PA levels and lower SB. That is, people who are satisfied with their look physically, have a good physical condition, report for motor tasks competence, present a capable strength in situations where strength is required, and, in turn, feel comfortable developing physical exercise or sports activities, they could report more intentions to be active people during out-of-school leisure time and in the future [11,25].

Lines 76-81; the introduction needs to do a better job of helping support these objectives. Specifically, it is unclear based on the literature why self-concept would predict PA and SB. Additionally, new concepts of intentions and out-of-school PA are newly introduced. In the objective 2, why is extrinsic motivation not explored as well? This needs to be clearer.

Response: Thanks again to the reviewer for this suggestion. First, we have added more information with the intention to support (based on the literature) why physical self-concept could be related to students’ physical activity levels and sedentary behavior.

“Conversely, individuals who report high levels of PSC could also present higher PA levels and lower SB. That is, people who are satisfied with their look physically, have a good physical condition, feel competent in tasks that require motor physical involvement and, in turn, feel comfortable being physically active, which may increase their intention to be active during out-of-school leisure time and in the future [11,25]”.

Sánchez-Miguel, P.A.; Leo, F.M.; Amado, D.; Pulido, J.J.; Sánchez-Oliva, D. Relationships between Physical Activity Levels, Self-Identity, Body Dissatisfaction and Motivation among Spanish High School Students. J. Hum. Kinet. 2017, 59, 29-38. https://doi.org/10.1515/hukin-2017-0145

Añez, E.; Fornieles-Deu, A.; Fauquet-Ars, J.; López-Guimerà, G.; Puntí-Vidal, J.; Sánchez-Carracedo, D. Body image dissatis-faction, physical activity and screen-time in Spanish adolescents. J. Health Psychol. 2018, 23, 36-47. https://doi.org/10.1177%2F1359105316664134

Second, regarding why dimensions of extrinsic motivation were not explored, we only measured the two motivational “extremes” such as intrinsic motivation and amotivation. However, we assume that the inclusion of extrinsic regulations in our study would have been an important addition to how physical self-concept is related to other students’ motivational regulations, their levels of physical activity and sedentary behavior. This is a limitation of our study that we must assume. We have added more information about this limitation in the specific section:

“Third, as the authors only measured intrinsic motivation and amotivation, it was not possible to know how external regulations may mediate between PSC and PA levels and SB. It would be interesting for future research including the entire spectrum of motivational regulations (also measuring extrinsic regulations) to better understand how PSC could be related to other students’ motivational regulations, their PA levels and SB”.

 Line 99; what is meant by conditioned classrooms?

Response: Sorry for this unclear term. We have change the sentence as follow:

“…Students’ who agreed to participate in the study completed paper-and-pencil questionnaires in their classrooms in a quiet environment and before a Physical Education lesson”.

Line 110; is this supposed to measure the Marsh model; if so why doesn’t it examine all the subcomponents.

Response: We used Marsh model to explain the global components of self-concept in the Introduction section. However, the current study analyzed the physical self-concept measured using the Physical Self-Perception Profile that contains five factors: fitness, motor/physical competence, appearance, physical strength, and self-esteem as previous studies have used (Sánchez-Miguel et al., 2017).

Moreno-Murcia, J.A.; Cervelló-Gimeno, E.; Vera-Lacárcel, J. A., Ruiz Pérez, L.M. Physical self-concept of Spanish schoolchildren: Differences by gender, sport practice and levels of sport involvement. J. Educ Hum. Develop. 2007, 1, 1-17.

Sánchez-Miguel, P.A.; Leo, F.M.; Amado, D.; Pulido, J.J.; Sánchez-Oliva, D. Relationships between Physical Activity Levels, Self-Identity, Body Dissatisfaction and Motivation among Spanish High School Students. J. Hum. Kinet. 2017, 59, 29-38. https://doi.org/10.1515/hukin-2017-0145

Lines 121-129; once again, why only these times of the BREQ-3?

Response: We totally agree with the reviewer that the inclusion of the extrinsic regulations it would have nurtured the work with more information on how physical self-concept is associated with the types of motivation (also extrinsic regulations) of the students and, in turn, with the intentions of being physically active and sedentary behavior. However, we did not include these regulations and we must assume as a limitation of our work.

“Despite the current research did not measure extrinsic regulations in line of a previous study [11], further investigations should examine how other students’ motivations towards PA could be associated with PSC and outcomes related to students’ PA levels and SB. For instance, practicing PA by aesthetic reasons could mediate the relationship between PSC factors (i.e., appearance, self-esteem or physical fitness among others) and students’ PA and SB levels”.

Lines 129-137; please try to show these items in the introduction.

Response: We have included these items in the Introduction section.

Data analysis; why did the authors create four models, it seems it could be just one model with each of the dependent variables all being together as dependent variables. This would help highlight what variables are of most importance and allow for a stronger discussion section to be written.

Response: We totally agree with the Reviewer that one model with each of the dependent variables could be also a good solution. However, we wanted to test in an independently way the direct effect between VI (i.e., students’ physical self-concept) on VD (i.e, PA variables and SB) as first goal of our study. Then, we wanted to analyze the possible mediating role of students’ intrinsic and students’ amotivation in this relation.

Figure 1 is very hard to read; can a higher quality figure be included?

Response: Sorry for not providing a quality figure in the last version of the manuscript. We have modified it.

Line 324-327; why was this the case; the authors really need to discuss at some point.

Response: We think that this comment has been previously attended. As we have commented, we have included the following information in the Discussion section:

“Third, as the authors only measured intrinsic motivation and amotivation, it was not possible to know how external regulations may mediate between PSC and PA levels and SB. It would be interesting for future research including the entire spectrum of motivational regulations (also measuring extrinsic regulations) to better understand how PSC could be related to other students’ motivational regulations, their PA levels and SB”.

Final question; were any analyses performed to see if there were sex/gender differences in the variables or fit of the model?

Response: The SEM was adjusted by students’ sex. For simplicity, these results were not shown, but we agree with the reviewer that they could be explained through the text. As preliminary analysis, an ANOVA (i.e., not represented in the text, we can include it as supplemental material if the reviewer considers necessary) was conducted including the students’ sex as a factor. We found significant differences in all dependent variables considering students’ sex. For this reason, we decided to include the sex as a covariate in our models. We have accordingly completed the Data analysis and Results sections.

Reviewer 2 Report

Although the article takes up an important aspect of young people's lives, which is PA, I get the impression that the authors are up to the task, which will be discussed below.

Abstract:
1. Do not use coma to big numbers like 1 003 and so throughout the text (l.15-17)
2. In the first appearance, please use the whole name and then (in the bracket) abbreviation, which should be used throughout text like: Physical self-concept (PSC).
Introduction:
1. Authors often refer to multiple studies without providing supporting references except for one (e.g., l. 29, 39, 41, 42, 51 and so on).
2. L. 40-41, Please replace the word 'posture' to 'position'
3. l.47 why semi-colon before 'or'?
4. If PSC is related to environment also, why Authors did not undertake into consideration extrinsic motivation e.g., peers'/friends', parental, teacher's support?
Material and Methods:
1. l.91 - '49.8' -
it is the percentage of girls calculated from the entire group of respondents, am I right? So where are the boys and their description?
2. 'school director' change for 'school principal' or 'head of school'
3. Don't you need parental permission to survey juveniles (under 18 years old)?
4. L. 106 - please provide references to the APA's ethical guidelines.
5. L. 107 - Were the responders informed about the voluntary nature of the survey?
6. L. 130 - Has the tool measuring Intentions to engage in PA been used before, or it is the Authors' original proposition?
7. L. 134-137 - to measure 'Out of school PA levels' Authors could and should used existing, well known, verified and validated tools, instead propose your own, solutions (contrary to the principle: 'you do not break down open doors').  
8. L. 143 - the same with the "Screen time" tool.

Considering the above, I consider the prepared manuscript not mature enough to be published in this form.
I think the authors should focus on the tools that should have been used in this research in order to be able to relate them to previous similar studies.

Please see
a useful bibliography below:
  1. J. F. Sallis, J.J. Prochaska, and W.C. Taylor, A review of correlates of physical activity of children and adolescents, Medicine & Science in Sports & Exercise 2000, 32, 963-975.
  2. J. F. Sallis, J.J. Prochaska, and W.C. Taylor, Correlates of physical activity in a national sample of girls and boys in grades 4 through 12, Health Psychology 1999, 18, 410-415.
  3. M. Belanger, M. Casey, M. Cormier, A.L. Filion, G. Martin, S. Aubut et. al, Maintenance and decline of physical activity during adolescence: Insights from a qualitative study”, International Journal of Behavioral Nutrition and Physical Activity 2001, 8 (117), retrieve from http://www.ijbnpa.org/content/8/1/117
  4. Prochaska, J.J.; Sallis, J.F.; Long, B. A physical activity screening measure for use with adolescents in primary care. Arch Pediatr Adolesc Med 2001, 155, 554-559, doi:10.1001/archpedi.155.5.554.
  5. Currie, C.; Gabhainn, S.; Godeau, E.; Roberts, C.; Smith, R.; Currie, D.; Picket, W.; Richter, M.; Morgan, A.; Barnekow V. Inequalities in young people ‘s health. HBSC international report from the 2005/2006 survey; World Health Organization: Copenhagen, Denmark, 2008.
  6. HBSC 2020 report. Available online: https://imid.med.pl/pl/aktualnosci/jakie-sa-polskie-nastolatki-raport-hbsc-2020 (accessed on 7 January 2021).
  7. https://journals.sagepub.com/doi/10.1177/1403494809105289
  8. https://pubmed.ncbi.nlm.nih.gov/29473720/
  9. WHO guidelines on physical activity and sedentary behaviour. Geneva: World Health Organization; 2020. License: CC BY-NC-SA 3.0 IGO, https://apps.who.int/iris/handle/10665/337001 (accessed on 26 November 2020).
  10. U.S. Department of Health and Human Services. Physical activity guidelines for Americans, 2nd ed.; U.S. Department of Health and Human Services: Washington, DC, 2018.
  11. World Health Organization. Global Recommendations on Physical Activity for Health. Available online: https://www.who.int/ncds/prevention/physical-activity/global-action-plan-2018-2030/en/ (accessed on 26 November 2020).
  12. World Health Organization, Every move counts towards better health – says WHO. Available online: https://www.who.int/news/item/25-11-2020-every-move-counts-towards-better-health-says-who (accessed on 25 November 2020).

Author Response

Reviewer 2

Although the article takes up an important aspect of young people's lives, which is PA, I get the impression that the authors are up to the task, which will be discussed below.

Response: We would like to thank to the Reviewer for the positive and valuable comments provided to our work. We have addressed all these recommendations point-by-point with the aim to improve the quality of the manuscript.

Abstract:
1. Do not use coma to big numbers like 1 003 and so throughout the text (l.15-17)

Response: Thanks for this objection. We have followed the reviewer’s recommendation.

  1. In the first appearance, please use the whole name and then (in the bracket) abbreviation, which should be used throughout text like: Physical self-concept (PSC).

Response: We have used this recommendation throughout text for physical self-concept (PSC).Introduction:

1. Authors often refer to multiple studies without providing supporting references except for one (e.g., l. 29, 39, 41, 42, 51 and so on).

Response: Thanks again for this recommendation. We have included more references when we refer to various studies. We have previously used references related to systematic reviews that included several studies related to the topic.

  1. L. 40-41, Please replace the word 'posture' to 'position'

Response: Thanks for this objection. We have used position instead of posture.

  1. l.47 why semi-colon before 'or'?

Response: We have changed semi-colon by a comma.

  1. If PSC is related to environment also, why Authors did not undertake into consideration extrinsic motivation e.g., peers'/friends', parental, teacher's support?

Response: We totally agree with the reviewer in this point. Grounded in self-determination theory, we only measured the two motivational “extremes” such as intrinsic motivation and amotivation. However, we assume that the inclusion of extrinsic regulations in our study would have been an important addition to how physical self-concept is related to other students’ motivational regulations, their levels of physical activity and sedentary behavior. This is a limitation of our study that we must assume. We have added more information about this limitation in the specific section:

“Third, as the authors only measured intrinsic motivation and amotivation, it was not possible to know how external regulations may mediate between PSC and PA levels and SB. It would be interesting for future research including the entire spectrum of motivational regulations (also measuring extrinsic regulations) to better understand how PSC could be related to other students’ motivational regulations, their PA levels and SB”.Material and Methods:

  1. l.91 - '49.8' - it is the percentage of girls calculated from the entire group of respondents, am I right? So where are the boys and their description?

Response: Sorry for not adding the boys’ percentage. We have included it as follow:

“The sample of the present study consisted of 1998 students (49.8% girls and 50.2% boys,…”

  1. 'school director' change for 'school principal' or 'head of school'

Response: Thanks again for this consideration. We have used “school principal”.

  1. Don't you need parental permission to survey juveniles (under 18 years old)?

Response: Yes, of course. We totally agree with the reviewer. The parental permission was asked to the students. We have specified this information in the Participants and Procedures section. Sorry for not including this information in the first version.

“After they agreed to participate in the study, the research assistants provided each student with a letter of information about the voluntary participation and a parental consent form. Students’ who agreed to participate in the study completed paper-and-pencil questionnaires in their classrooms in a quiet environment and before a Physical Education lesson”.

  1. L. 106 - please provide references to the APA's ethical guidelines.

Response: We have included the APA’s reference:

American Psychological Association. Publication Manual of the American Psychological Association (7th ed.). 2019; Washington, DC: American Psychological Association.

  1. L. 107 - Were the responders informed about the voluntary nature of the survey?

Response: Yes, they were informed that their participation in the study was voluntary. We apologize for not including this information in the first version.

“After they agreed to participate in the study, the research assistants provided each student with a letter of information about the voluntary participation and a parental consent form. Students’ who agreed to participate in the study completed paper-and-pencil questionnaires in their classrooms in a quiet environment and before a Physical Education lesson”.

  1. L. 130 - Has the tool measuring Intentions to engage in PA been used before, or it is the Authors' original proposition?

Response: Yes, this scale has been previously used by Sánchez-Miguel et al. (2017). We have included this reference in the explanation of this tool, specifically, in the Instruments section.

Sánchez-Miguel, P.A.; Leo, F.M.; Amado, D.; Pulido, J.J.; Sánchez-Oliva, D. Relationships between Physical Activity Levels, Self-Identity, Body Dissatisfaction and Motivation among Spanish High School Students. J. Hum. Kinet. 2017, 59, 29-38. https://doi.org/10.1515/hukin-2017-0145

  1. L. 134-137 - to measure 'Out of school PA levels' Authors could and should use existing, well known, verified and validated tools, instead propose your own, solutions (contrary to the principle: 'you do not break down open doors').  
  2. L. 143 - the same with the "Screen time" tool.

Considering the above, I consider the prepared manuscript not mature enough to be published in this form.
I think the authors should focus on the tools that should have been used in this research in order to be able to relate them to previous similar studies.

Response: We answer these two questions together. We totally agree with the reviewer. We have assumed these points together and as important issues of our work. However, “screen time” tool was previously used by Sánchez-Miguel et al. (2017). We know that recent tools, for example the YLSB developed by Cabanas-Sánchez et al. (2018), have been designed to assess the screen time and SB. However, the current work was developed before publication of Cabanas-Sánchez et al.’s work. To our knowledge, other studies have also used self-reported questionnaires to measure the screen time. Therefore, we honestly think that does not detract the credibility of our findings, as there is previous research that has followed a similar method to assess sedentary screen time and, therefore, this reinforces our idea (see references below). But we must assume as an important limitation of our work.

Katzmarzyk, P.T., Staiano, A.E. Relationship between meeting 24-hour movement guidelines and cardiometabolic risk factors in children. J Phys Act Health. 2017, 14(10), 779-784.

Roberts, K.C., Yao, X., Carson, V., Chaput, J. P., Janssen, I., Tremblay, M.S. Meeting the Canadian 24-hour movement guidelines for children and youth. Health Rep. 2017, 28(10), 3-7.

Roman-Viñas, B., Chaput, J.P., Katzmarzyk, P.T., Fogelholm, M., Lambert, E.V., Maher, C., ... Tremblay, M.S. Proportion of children meeting recommendations for 24-hour movement guidelines and associations with adiposity in a 12-country study. Int. J. Behav. Nutr. Phys. Act. 2016, 13, 1-10.

Sánchez-Miguel, P.A.; Leo, F.M.; Amado, D.; Pulido, J.J.; Sánchez-Oliva, D. Relationships between Physical Activity Levels, Self-Identity, Body Dissatisfaction and Motivation among Spanish High School Students. J. Hum. Kinet. 2017, 59, 29-38. https://doi.org/10.1515/hukin-2017-0145

“Finally, we have used an own created scale for the current study to assess out of school PA levels instead of using other previous validated instruments [51].”

Migueles, J.H.; Aadland, E.; Andersen, L.B.; Brønd, J.C.; Chastin, S.F.; Hansen, B.H.; Ortega, F.B. GRANADA consensus on analytical approaches to assess associations with accelerometer-determined physical behaviours (physical activity, sedentary behaviour and sleep) in epidemiological studies. Br. J. Sports Med. 2021, 1-9. https://doi.org/10.1136/bjsports-2020-103604Please see a useful bibliography below:

Response: Thanks again. We have followed this recommendation provided by the Reviewer. We have included some of these works in our current version of the manuscript.

  1. J. F. Sallis, J.J. Prochaska, and W.C. Taylor, A review of correlates of physical activity of children and adolescents, Medicine & Science in Sports & Exercise 2000, 32, 963-975.
  2. J. F. Sallis, J.J. Prochaska, and W.C. Taylor, Correlates of physical activity in a national sample of girls and boys in grades 4 through 12, Health Psychology 199918, 410-415.
  3. M. Belanger, M. Casey, M. Cormier, A.L. Filion, G. Martin, S. Aubut et. al, Maintenance and decline of physical activity during adolescence: Insights from a qualitative study”, International Journal of Behavioral Nutrition and Physical Activity 20018 (117), retrieve from http://www.ijbnpa.org/content/8/1/117
  4. Prochaska, J.J.; Sallis, J.F.; Long, B. A physical activity screening measure for use with adolescents in primary care. Arch Pediatr Adolesc Med 2001155, 554-559, doi:10.1001/archpedi.155.5.554.
  5. Currie, C.; Gabhainn, S.; Godeau, E.; Roberts, C.; Smith, R.; Currie, D.; Picket, W.; Richter, M.; Morgan, A.; Barnekow V. Inequalities in young people ‘s health. HBSC international report from the 2005/2006 survey; World Health Organization: Copenhagen, Denmark, 2008.
  6. HBSC 2020 report. Available online: https://imid.med.pl/pl/aktualnosci/jakie-sa-polskie-nastolatki-raport-hbsc-2020 (accessed on 7 January 2021).
  7. https://journals.sagepub.com/doi/10.1177/1403494809105289
  8. https://pubmed.ncbi.nlm.nih.gov/29473720/
  9. WHO guidelines on physical activity and sedentary behaviour. Geneva: World Health Organization; 2020. License: CC BY-NC-SA 3.0 IGO, https://apps.who.int/iris/handle/10665/337001 (accessed on 26 November 2020).
  10. U.S. Department of Health and Human Services. Physical activity guidelines for Americans, 2nd ed.; U.S. Department of Health and Human Services: Washington, DC, 2018.
  11. World Health Organization. Global Recommendations on Physical Activity for Health. Available online: https://www.who.int/ncds/prevention/physical-activity/global-action-plan-2018-2030/en/ (accessed on 26 November 2020).
  12. World Health Organization, Every move counts towards better health – says WHO. Available online: https://www.who.int/news/item/25-11-2020-every-move-counts-towards-better-health-says-who (accessed on 25 November 2020).

Reviewer 3 Report

This manuscript is a report on a cross-sectional study in which associations among physical self-concept, motivation, physical activity, and sedentary behavior were examined in a sample of Spanish secondary school students. The topic falls squarely within the domain covered by this journal. The manuscript is very well-written, and the large sample size and strong theoretical foundation are desirable features of the study presented in the manuscript. These positive impressions notwithstanding, I have several concerns about the manuscript in its current form:

  1. It should be specified that hard copy (or paper-and-pencil) rather than online questionnaires were administered.

  1. Was parental consent required/obtained? If so, it should be indicated on l. 98 and l. 105.

  1. On l. 126, it might make sense to replace “a good” with “adequate.”

  1. With regard to the sentence on l. 147-148, have the single-item scales described in sections 2.3.4-2.3.6 been validated against objective measures of the constructs they are intended to assess? Such validation is necessary to have confidence in the findings of the current study.

  1. Given the nonexperimental nature of the research design in the current study, references to “independent” and “dependent” variables on l. 174 and l. 176 should be replaced with “predictor” and “criterion,” respectively.

  1. On l. 230, it should be “design” instead of “desing.”

  1. The authors have done a great job of refraining from using language implying the existence causal relationships, but evidence that “physical self-concept improves perceived competence” (l. 231-232) is primarily correlational rather than experimental.

  1. On l. 245, it might make sense to replace “conducted” with “proposed” or “evaluated.”

  1. The recommendation on l. 237-238 to conduct “more experimental studies” is especially important given that other (including reciprocal) relationships among the variables in the model are possible and, indeed, likely, a point that should be recognized in the Discussion.

  1. On l. 237, it should be “these findings should.”

Author Response

Reviewer 3

This manuscript is a report on a cross-sectional study in which associations among physical self-concept, motivation, physical activity, and sedentary behavior were examined in a sample of Spanish secondary school students. The topic falls squarely within the domain covered by this journal. The manuscript is very well-written, and the large sample size and strong theoretical foundation are desirable features of the study presented in the manuscript. These positive impressions notwithstanding, I have several concerns about the manuscript in its current form:

Response: We would like to thank to the Reviewer for the positive and valuable comments provided to our work. We have addressed all these recommendations point-by-point (marked in red color) with the aim to improve the quality of the manuscript.

  1. It should be specified that hard copy (or paper-and-pencil) rather than online questionnaires were administered.

Response: Thanks for this recommendation. We have included in the Participants and Procedures subsection the following information:

“Participants completed paper-and-pencil questionnaires in their classrooms in a quiet environment and before a Physical Education lesson”.

  1. Was parental consent required/obtained? If so, it should be indicated on l. 98 and l. 105.

Response: We have also added this information in the Participants and Procedures section:

“After they agreed to participate in the study, the research assistants provided each student with a letter of information about the voluntary participation and a parental consent form. Students’ who agreed to participate in the study completed paper-and-pencil questionnaires in their classrooms in a quiet environment and before a Physical Education lesson”.

  1. On l. 126, it might make sense to replace “a good” with “adequate.”

Response: Thanks. Replaced.

  1. With regard to the sentence on l. 147-148, have the single-item scales described in sections 2.3.4-2.3.6 been validated against objective measures of the constructs they are intended to assess? Such validation is necessary to have confidence in the findings of the current study.

Response: We understand the reviewer objection. However, we cannot address the validity of a scale composed by one item. We cited some works which have previously used single-item instruments. We have recognized this point as a limitation of our work.

“Also, some instruments used in the current work present a single-item structure (e.g., screen time), therefore, it is not possible to test the validity and reliability of these instruments”.

We know that recent tools, for example the YLSB developed by Cabanas-Sánchez et al. (2018), have been designed to assess the screen time and SB. However, the current work was developed before publication of Cabanas-Sánchez et al.’s work. To our knowledge, other studies have also used self-reported questionnaires to measure the screen time. Therefore, we honestly think that does not detract the credibility of our findings, as there is previous research that has followed a similar method to assess sedentary screen time and, therefore, this reinforces our idea (see references below). But we must assume as an important limitation of our work.

Katzmarzyk, P.T., Staiano, A.E. Relationship between meeting 24-hour movement guidelines and cardiometabolic risk factors in children. J Phys Act Health. 2017, 14(10), 779-784.

Roberts, K.C., Yao, X., Carson, V., Chaput, J. P., Janssen, I., Tremblay, M.S. Meeting the Canadian 24-hour movement guidelines for children and youth. Health Rep. 2017, 28(10), 3-7.

Roman-Viñas, B., Chaput, J.P., Katzmarzyk, P.T., Fogelholm, M., Lambert, E.V., Maher, C., ... Tremblay, M.S. Proportion of children meeting recommendations for 24-hour movement guidelines and associations with adiposity in a 12-country study. Int. J. Behav. Nutr. Phys. Act. 2016, 13, 1-10.

Sánchez-Miguel, P.A.; Leo, F.M.; Amado, D.; Pulido, J.J.; Sánchez-Oliva, D. Relationships between Physical Activity Levels, Self-Identity, Body Dissatisfaction and Motivation among Spanish High School Students. J. Hum. Kinet. 2017, 59, 29-38. https://doi.org/10.1515/hukin-2017-0145

  1. Given the nonexperimental nature of the research design in the current study, references to “independent” and “dependent” variables on l. 174 and l. 176 should be replaced with “predictor” and “criterion,” respectively.

Response: Thank you so much again. We have followed the reviewer’s suggestion.

  1. On l. 230, it should be “design” instead of “desing.”

Response: Sorry for this mistake. We have corrected it.

  1. The authors have done a great job of refraining from using language implying the existence causal relationships, but evidence that “physical self-concept improves perceived competence” (l. 231-232) is primarily correlational rather than experimental.

Response: Thanks again. We have smoothed out the possible relationship between physical self-concept and perceived competence in adolescence.

  1. On l. 245, it might make sense to replace “conducted” with “proposed” or “evaluated.”

Response: Modified.

  1. The recommendation on l. 237-238 to conduct “more experimental studies” is especially important given that other (including reciprocal) relationships among the variables in the model are possible and, indeed, likely, a point that should be recognized in the Discussion.

Response: Thank you very much for your comment. The authors agree with the Reviewer 3, and we have therefore highlighted this aspect as a future perspective.

“Given the cross-sectional nature of the study, future experimental studies would allow us to examine changes in the identified relationships over time and provide more robust evidence of the relationship between PSC, motivational processes, and active or SB.”

  1. On l. 237, it should be “these findings should.”

Response: Modified.
